# An Adaptive Alternating Magnetic Interference Suppression (AAIS) Algorithm for Geomagnetic Vector Measurement

**DOI:** 10.3390/s22103642

**Published:** 2022-05-10

**Authors:** Weilin Wang, Keyan Li, Zhihao Yang, Jun Chen, Linliang Miao, Jun Ouyang, Xiaofei Yang

**Affiliations:** School of Optical and Electronic Information, Huazhong University of Science and Technology, Wuhan 430074, China; wwl@hust.edu.cn (W.W.); keyanli@hust.edu.cn (K.L.); zhihao_yang@hust.edu.cn (Z.Y.); chen_jun@hust.edu.cn (J.C.); mll@hust.edu.cn (L.M.); yangxiaofei@hust.edu.cn (X.Y.)

**Keywords:** alternating magnetic interference, electrical equipment, geomagnetic vector measurement

## Abstract

To achieve high-precision vector measurement values in a geomagnetic field, it is necessary to develop methods for overcoming alternating magnetic interference (AMI), which is generated by electrical equipment. This paper proposes the adaptive alternating magnetic interference suppression (AAIS) algorithm. In this algorithm, first, only a triaxial fluxgate sensor measures the magnetic field data. The time–frequency diagram of the total magnetic field is obtained quickly through short-time Fourier transform and wavelet transform. Additionally, the time and frequency of AMI appearance are analyzed. Then, the triaxial adaptive notch filter suppresses the three-component related magnetic interference. Herein, simulation and actual experiments are performed to verify the effectiveness of AAIS. The results indicate that the algorithm can quickly detect the frequencies of AMI from the total magnetic field and adaptively fit their amplitude and phase on the vector magnetic field. Finally, AAIS can suppress the interference effectively. The AAIS algorithm realizes error compensation for the vector measurement values by the total magnetic field, which effectively improves the vector measurement accuracy of the geomagnetic field. We highlight that the AAIS algorithm is effective for AMIs of different frequencies, numbers, and intensities without reference sensors. Our work has practical implications in airborne, vehicle-mounted, and shipborne geomagnetic vector detection.

## 1. Introduction

A triaxial magnetometer fixed on a plane can measure a vector magnetic field and provide a wealth of information, and thus, it has extensive applications in the fields of geomagnetic navigation, magnetic anomaly detection, and nondestructive testing [1,2,3]. For vector geomagnetic measurement systems, the measurement accuracy of the magnetic sensor on a plane is affected by the ferromagnetic materials [4] and electrical equipment on the plane [5]. The Tolles–Lawson model divides the first kind of error into permanent, induced, and eddy-current interferential fields, which form the T–L model [6,7]. The solution of the compensation coefficient for this model has been extensively studied [8,9,10,11]. However, the T–L model is not valid for alternating magnetic interference (AMI) generated by the electrical equipment on the plane. In practice, there are many other interference sources that can generate AMI, such as propellers and interphones. Their signal characteristics are narrowband noise [12].

Primarily, there are two methods to suppress AMI. The first is placing the magnetometer far enough from the platform magnetic source [13], which belongs to hardware compensation. Mounting methods include support bar [14,15,16,17,18,19] and rope [20,21]. The distance of the magnetometer from the noise source varies with the type of carrier, with a typical reference value of 3 m given by Walter and Braun [13,22]. Such modifications increase the size of the aeromagnetic measurement system, and seriously affect the stability of the aircraft. The second is software compensation, including the synchronous reference subtraction method and the coherent noise suppression (CNS) method. The synchronous reference subtraction method is a direct denoising method in the time domain that subtracts the signals from two measurement values located some distance away to remove the common-mode geomagnetic noise [23,24]. However, the amplitudes of AMI measured by the reference sensor and the target sensor are different, so it is difficult to eliminate this interference. Second, Chen et al. [25] proposed the coherent noise suppression (CNS) method, which analyzes the correlation of AMI on the target sensor with the reference sensor in the frequency domain [26]. The effectiveness of this method is only verified by using two optical pump magnetometers to suppress AMI on the total geomagnetic measurement values. At present, these compensation methods are mostly used for the compensation of geomagnetic total field measurement values.

For a dynamic measurement system, the vector measurement values of a triaxial magnetometer are sensitive to the installation attitude and the maneuver caused by the plane, which is different from the total geomagnetic measurement values [27,28,29]. On the one hand, it is difficult to achieve complete alignment of the target sensor and the reference sensor for each axis [30]; it is, therefore, impossible for the two sensors to suppress AMI by directly subtracting effectively. On the other hand, the vector measurement values are a mixture of valuable information caused by plane maneuvering and interference information caused by AMI [31]. Maneuvering can reflect the attitude angle of the plane [32], so it is necessary to measure and extract this information. However, it is strongly correlated with AMI on the reference sensor and the target sensor [33]. The CNS method will equally eliminate the two kinds of information, which is unacceptable for the vector measurement system of the geomagnetic field. There is still no compensation method suitable for suppressing AMI on vector geomagnetic measurement values.

This paper introduces the adaptive alternating magnetic interference suppression (AAIS) algorithm. It can realize vector compensation of the triaxial magnetometer without reference sensors. The remainder of the paper is organized as follows. Section 2 presents the theoretical model of AAIS. In Section 3, simulations are performed to evaluate the method. In Section 4, the experimental results are presented. In Section 5, the effectiveness of AAIS is discussed in different situations. Finally, we conclude in Section 6.

## 2. The AAIS Algorithm

The AAIS algorithm realizes the vector compensation of the geomagnetic measurement values by the measurement values of the total magnetic field. It extracts the frequency of the AMI from the total magnetic field value and uses it to implement interference compensation of the vector measurements.

### 2.1. Theoretical Model

The magnetometer frame (M-frame) on the plane is defined as shown in Figure 1.

The origin of the M-frame is the center of the magnetometer’s sensitive axes, and the x-, y- and z-axes correspond to the right, forward, and vertical directions of the plane, respectively. The M-frame is a right-handed Cartesian coordinate system., $H_{x}$, $H_{y}$ and $H_{z}$ are the three components of the magnetometer measurement values.

Assume that $\alpha_{0}$, $\beta_{0}$ and $\gamma_{0}$ are the direction cosine angles of the plane, and $H_{n}$ is the total geomagnetic reference field. $H_{x}$, $H_{y}$, and $H_{z}$ are the three components of the magnetometer in the M-frame, which can be expressed as Equation (1).
(1)$$\left\{\begin{array}{c} H_{x}=H_{n}\cos(\alpha_{0})\\ H_{y}=H_{n}\cos(\beta_{0})\\ H_{z}=H_{n}\cos(\gamma_{0}) \end{array}\right.$$

The magnetic field changes caused by $\alpha_{0}$, $\beta_{0}$, and $\gamma_{0}$ are called the maneuver of the plane. The maneuvering information measured by the vector magnetometer is combined with inertial sensors to calculate the attitude angles of the plane; the maneuvering information measured by a vector magnetometer is, therefore, a valuable signal we need to measure and extract.

Assume the frequency of interference generated by electrical equipment is ω. The vector noise $N_{x}$, $N_{y}$, $N_{z}$ can be expressed as Equation (2).
(2)$$\left\{\begin{array}{c} N_{x}=A_{1}\cos(\omega t+\varphi_{1})\\ N_{y}=A_{2}\cos(\omega t+\varphi_{2})\\ N_{z}=A_{3}\cos(\omega t+\varphi_{3}) \end{array}\right.$$

The vector measurement values of the triaxial magnetometer $B_{x}$, $B_{y}$ and $B_{z}$ can be expressed as Equation (3).
(3)$$\left\{\begin{array}{c} B_{x}=N_{x}+H_{x}=A_{1}\cos(\omega t+\varphi_{1})+H_{n}\cos(\alpha_{0})\\ B_{y}=N_{y}+H_{y}=A_{2}\cos(\omega t+\varphi_{2})+H_{n}\cos(\beta_{0})\\ B_{z}=N_{z}+H_{z}=A_{3}\cos(\omega t+\varphi_{3})+H_{n}\cos(\gamma_{0}) \end{array}\right.$$

The total measurement values of the triaxial magnetometer $B_{m}$ can be calculated by Equation (4).
(4)$$B_{m}={\left(B_{x}^{2}+B_{y}^{2}+B_{z}^{2}\right)}^{1/2}$$

Substituting Equation (3) into Equation (4).
(5)$$\begin{array}{l} B_{m}(t)=[F_{1}\cos^{2}(\omega t)+F_{2}\sin^{2}(\omega t)+F_{3}\sin(2\omega t)+2A_{1}B_{1}\cos(\omega t+\varphi_{1})\\ +2A_{2}B_{2}\cos(\omega t+\varphi_{2})+2A_{3}B_{3}\cos(\omega t+\varphi_{3})+H_{n}^{2}{]}^{\frac{1}{2}} \end{array}$$
where $F_{1}$, $F_{2}$ and $F_{3}$ are shown in Equation (6).
(6)$$\left\{\begin{array}{l} F_{1}=A_{1}^{2}\cos^{2}\varphi_{1}+A_{2}^{2}\cos^{2}\varphi_{2}+A_{3}^{2}\cos^{2}\varphi_{3}\\ F_{2}=A_{1}^{2}\sin^{2}\varphi_{1}+A_{2}^{2}\sin^{2}\varphi_{2}+A_{3}^{2}\sin^{2}\varphi_{3}\\ F_{3}=-\frac{1}{2}(A_{1}^{2}\cos2\varphi_{1}+A_{2}^{2}\cos2\varphi_{2}+A_{3}^{2}\sin2\varphi_{3}) \end{array}\right.$$

Comparing Equations (3) and (5), the components $B_{x}$, $B_{y}$ and $B_{z}$ compose the frequency ω of AMI and maneuver information caused by $\alpha_{0}$, $\beta_{0}$ and $\gamma_{0}$. However, the total magnetic field only contains AMI information, without the maneuver information. If the interference frequency is accurately extracted from the total magnetic field, the corresponding noise on the components can be suppressed. Then, the high-precision measurement values of the geomagnetic vectors, including the maneuver information, can be obtained.

In the actual measurement system, the interference of multiple frequencies exists simultaneously, and the amplitude and phase of each frequency are random. Therefore, it is necessary to realize the adaptive detection of multiple interference sources to eliminate AMI.

### 2.2. Algorithm Design

The principal diagram of the AAIS algorithm is shown in Figure 2. We only use one triaxial fluxgate sensor to collect the magnetic field vector data $B_{xm}$, $B_{ym}$, $B_{zm}$. First, calculate the total magnetic field from the measured values of the triaxial magnetometer. Then, through short-time Fourier transform (STFT) and wavelet transform (WT) analysis, the time–frequency diagram of the total magnetic field is obtained. The STFT is used to observe the frequency components of the magnetic field in a fixed time window. Determine whether a new frequency of AMI appears. (1) If yes, the WT will be used to find an accuracy time when it occurs, and the adaptive triaxial notch filter will be used to estimate and eliminate the corresponding interference. Finally, we obtain geomagnetic field vector measurement values without interference ($B_{x}$, $B_{y}$ and $B_{z}$). (2) If not, the signal will be directly output.

The Fast Fourier Transformation (FFT) is used to process a section of observed window. Based on the frequency spectrum, the amplitude and frequency information are detected, then fed into fuzzy clustering methods (FCM) to identify whether new frequencies exist. When AMI occurs, the wavelet transform is used to find the accuracy position of the mutation point in the observed window. The new frequency point detection technique used in this paper is referred to as FFT + FCM+ wavelet. It is realtively mature, as shown in reference [34].

For the time consumption of the algorithm, FFT + FCM+ wavelet has the characteristics of quick detection. The computer used in the experiment has an intel i7-8750H CPU and 8G RAM, with MATLAB software installed. Assuming that the sampling rate is 500 Hz, the sampling duration is 6 s, and the AMI occurs at 5.9 s. During a detection window of 0.2 s, the FFT takes 1.2 ms, the wavelet transform takes 49 ms, and the FCM takes 8 ms. There are three methods to detect the AMI, (i) FFT + FCM, which takes 276 s; (ii) wavelet + FCM, which takes 1.71 s; and (iii) FFT + FCM+ wavelet, which takes 325 s. For the FFT + FCM method, it takes the least time but it can only determine the presence of AMI between 5.8 s and 6 s and cannot determine the exact time. In contrast, Methods 2 and 3 are both capable of detecting specific times, when AMI occurs. Compared with Method 2, Method 3 has an 81% reduction in time, so, it has the features of quick detection and was applied in this paper. 

Notably, the frequency of AMI may contain multiple frequency components, ω,i=1,2,…n. Moreover, the triaxial adaptive notch filter suppresses the three components related to AMI, and a schematic diagram is shown in Figure 3. The meaning of the characters is shown in Table 1.

There are two parameters (amplitude *A* and phase *φ*) to be fitted, so the adaptive filter contains two weights, $h_{x0}$ and $h_{x1}$. To use sin and cos functions as a set of orthogonal basis functions, we add a phase shifter (90°). The orthogonal reference input is expressed as Equation (7).
(7)$$\left\{\begin{array}{c} x_{0k}=A\cos(k\omega_{i}+\varphi )\\ x_{1k}=A\sin(k\omega_{i}+\varphi ) \end{array}\right.$$

The update method of weight coefficients $h_{x0}$ and $h_{x1}$ are shown in Equation (8).
(8)$$\left\{\begin{array}{c} h_{0}(k+1)=h_{0}(k)+\mu e(k)x_{0}(k)\\ h_{1}(k+1)=h_{1}(k)+\mu e(k)x_{1}(k) \end{array}\right.$$

When e(k) reaches the minimum, the weight coefficients $h_{x0}$ and $h_{x1}$ no longer change. The transfer function of the adaptive notch filter is expressed as Equation (9).
(9)$$H(\omega )=\frac{\omega^{2}-2\omega \cos\omega_{i}+1}{\omega^{2}-2(1-\mu A^{2})\omega \cos\omega_{i}+1-2\mu A^{2}}$$

In summary, the AAIS algorithm has four characteristics: (1) without reference sensors; (2) quickly and accurately detect the frequency of AMI; (3) adaptively solve the amplitude and phase of AMI; and (4) effectively suppress AMI.

## 3. Simulations and Results

### 3.1. AAIS Suppression Effect on Static AMI

Assume that the geomagnetic environment is [28,000, 38,000, 10,000] nT, and the two frequencies of AMI, $f_{1}$ and $f_{2}$, are 75 Hz and 40 Hz, respectively. The AMI occurs at 0.3 s. The measurement noise of the magnetometer for each axis is 10 pT. The sampling rate is 500 Hz. The sampling time is 6 s. Considering the influence of the electronic circuit, the measured amplitude and phase on the three axes of the triaxial magnetometer may be different. Therefore, the AMIs are set on vectors as follows.
(10)$$\left\{\begin{array}{l} N_{x}=600(\sin(2\pi f_{1}t)+\sin(2\pi f_{2}t))\\ N_{y}=200(\sin(2\pi f_{1}t+\pi /2)+\sin(2\pi f_{2}t+\pi /2))\\ N_{z}=100(\sin(2\pi f_{1}t+\pi )+\sin(2\pi f_{2}t+\pi )) \end{array}\right.$$

The changed attitude angles simulate the maneuvering of the plane, which is shown in Table 2.

Figure 4 shows the simulation results. The magnetic field change caused by the plane maneuver is shown as blue curves. When the plane maneuver and AMI coexist, the magnetic field measurement values of the triaxial magnetometer are the orange curve with a −3000 nT offset for clarity. The data after AAIS are the yellow curve, which is offset by −6000 nT for clarity. For orange curves, the useful maneuvering information on the components is drowned by AMI, and we cannot recognize the actual movement state of the plane. For the yellow curve processed by AAIS, the AMI is suppressed, and the maneuver information is retained.

The time–frequency diagrams before and after AAIS are shown in Figure 5. In Figure 5a, we can see that the total magnetic field only contains the two frequencies of AMI, 40 Hz and 75 Hz. In Figure 5b–d, we can see that the frequencies of AMI and the plane maneuver coexist. After applying the AAIS, the AMI in the total and vector magnetic fields is eliminated. Simultaneously, the maneuver information on the three components is retained, which is as expected. The maneuvering frequencies of pitch, roll and yaw are 15 Hz, 80 Hz and 50 Hz, respectively. They are all reserved on the three components after being processed by AAIS. Therefore, the effectiveness of AAIS is not associated with the relationship between AMI frequencies and plane maneuver frequencies, which cannot be achieved with a single low-pass or high-pass filter.

The simulation results indicate that the AAIS algorithm detects the AMI frequencies automatically and quickly. It adaptively estimates the amplitude and phase of the interference. Finally, it suppresses AMI and retains useful maneuvering information about the three components. Despite the existence of AMI, the AAIS algorithm can accurately extract plane maneuvering information with the triaxial magnetometer.

### 3.2. AAIS Suppression Effect on Dynamic AMI

Due to the closed-loop control structure of the AAIS algorithm, it is not only good at suppressing AMI in the static case, but it is equally effective for AMI in the dynamic case. To illustrate this point, we have added dynamic experiments for magnetic noise.

In the dynamic AMI noise suppression experiment, the sampling rate is 500 Hz and the sampling duration is 24 s. The AMIs are divided into two parts: static and dynamic, denoted as AMI_s_ and AMI_d,_ separately. For AMI_s_, the frequency is 40 Hz, the magnitudes in the x-, y-, and z-axes are (600, 200, 100), and the phases are (0, Pi/2, Pi). The AMI_d_ changes every 6 s and is separately named AMI_d1_, AMI_d2_, AMI_d3_, AMI_d4_, as shown as in Table 3. The frequencies are different for the AMI_d1_ and AMI_d2_: the AMI_d2_ and AMI_d3_ are different in amplitude, whereas, the AMI_d3_ and AMI_d4_ are different in phase.

The results before and after AAIS are shown in Figure 6. The AAIS algorithm detects the changes in AMI at 0 s, 6 s, 12 s and 18 s. It then adjusts the compensation parameters adaptively. The time–frequency diagrams before and after the AAIS are shown in Figure 7. It can be seen that even though the frequencies, amplitudes and phases of AMIs have changed, the AAIS algorithm still achieves a good suppression effect.

The quantitative estimation of noise after AAIS was taken over a 1 s window and the results are shown in Table 4. It can be seen that the compensated AMI noises reach very low levels. Therefore, the AAIS algorithm is not only applicable to static AMI, but also still has good suppression effect on dynamic AMI.

## 4. Experiments and Results

There are many interference sources, such as motors, propellers, and interphones, that can generate AMI on actual flight vehicles. Although the frequency, amplitude, and phase of AMIs generated by different noise sources differ, they are all narrowband noise. In this paper, we carried out experiments based on the leakage AMI noise of the motor.

Depending on the type of power supply, motors can be divided into two categories: DC motors and AC motors. Aircraft can only be powered by DC batteries, so most of the motors used on a plane are DC motors. According to the presence or absence of a commutator, DC motors are divided into brushed and brushless. For both, a periodically rotating magnetic field is generated when it is running. However, the AAIS algorithm is not sensitive to the generation mechanism of the rotating magnetic field, but only to the frequency, amplitude, and phase of the AMI. Hence, the compensation experiments based on brushless DC motors are also applicable to other types of motors.

In addition, the effectiveness of the AAIS algorithm with brushless DC motors was verified with static and dynamic experiments. When the plane is stationary, the AAIS algorithm’s suppression of AMI is called static performance. When the attitude angles of the plane change, its suppression of AMI is called dynamic performance.

### 4.1. Experimental Design

The experimental system includes a triaxial fluxgate, DC brushless motor and data acquisition equipment, shown in Figure 8. The measurement noise of the fluxgate is 6 pTrms/√Hz @ 1 Hz. The sampling rate is 1024 Hz. They are all fixed on the nonmagnetic platform, which is made of epoxy resin. The platform is rotatable in three-dimensional space. The distance between the motor and the fluxgate is approximately 10 cm.

### 4.2. Static Performance

The relative position of the motor and the fluxgate sensor is fixed, and the plane is static. First, we measured the geomagnetic background signal for 60 s. We then turned on the motor for 180 s. Next, we turned off the motor and continuously measured the geomagnetic background signal for 230 s. The X, Y and Z components before and after AAIS are shown in Figure 9. The data processed by AAIS are offset by 20 nT for clarity.

We can see that the AAIS detects the interference frequencies successfully when the motor is running. It then estimates the interference amplitude and phases adaptively within 2 s. Finally, it eliminates the interference in the X, Y, Z components. When the motor was running stably and the geomagnetic background was constant, we quantified the compensation effect of the AAIS with the data with a time interval of 10 s (70 s–80 s). As shown in Table 5, where the IR is the improvement ratio, the standard deviation (STD) of the X component reduced from 3.63 nT to 0.42 nT, which is a reduction of 88.4%. The STD of the Y component decreased from 1.83 nT to 0.6 nT, a percentage of 67.2%. Finally, the STD of the Z component decreased from 7.31 nT to 0.54 nT, a reduction of 92.6%. These results fully confirm the effectiveness of the algorithm when the plane is static.

### 4.3. Dynamic Performance

In the dynamic experiment, the relative position of the motor and the fluxgate sensor were fixed. The nonmagnetic platform was randomly shaken to imitate the maneuvering changes of the plane. First, we measured the geomagnetic background for the period 0–30 s. Next, we shook the nonmagnetic platform at a small angle of ± 3° for 160 s. We then turned on the motor for 390 s. We continue to randomly shake the nonmagnetic plane for 510 s. Finally, we measured the geomagnetic background for 600 s. The measurements of the triaxial magnetometer and the result processed by AAIS are shown in Figure 10. The data processed by AAIS are offset by 300 nT for clarity.

The frequency spectrum of the data measured by the magnetometer and processed by the AAIS algorithm when the maneuver and AMI coexist (160 s to 390 s) is shown in Figure 11. This result indicates that AAIS detected four frequencies of AMI: 1.30 Hz, 2.20 Hz, 4.80 Hz, and 6.70 Hz. In Figure 11a, the total magnetic field contains four frequencies of AMI but without maneuver frequencies. The AAIS then suppresses the AMI of the four frequencies on three components, as shown in Figure 11b–d. The results demonstrate that AAIS suppressed the AMIs excellently and preserved the information of the platform maneuver effectively.

We quantified the compensation effect of the AAIS algorithm with the data from 398 s to 400 s, as shown in Table 6. The STD of the x component decreased from 7891.45 nT to 50.38 nT, which is a reduction of 99.4%. The STD of the y component decreased from 681.09 nT to 7.23 nT, a reduction of 98.9%. Finally, the STD of the z component decreased from 2062.7 nT to 14.10 nT, a percentage of 99.3%.

The results strongly indicate that the AAIS algorithm could effectively detect and eliminate AMI in a dynamic environment. It extracts useful geomagnetic information from complex AMI_s_, which reflects the attitude changes in the platform. The measurement accuracy of the geomagnetic vector is, therefore, effectively improved.

## 5. Discussion

To study the relationship between the effectiveness of AAIS and the dynamic AMI, when they are placed at different orientations, different rotation speeds, with different numbers, and at different distances. Based on the aforementioned static experiments, we performed extensive extended experiments. First, we placed the triaxial magnetometer at different orientations to the motor, such as at the east, south, west, and north. We then quantified the STD of vector geomagnetic measurements before and after the AAIS within an appropriate 2 s time period. The results are shown in Table 7.

The triaxial magnetometer was then placed at distances of 5 cm, 10 cm and 20 cm from the motor. The STDs of vector geomagnetic measurements before and after AAIS within an appropriate 2 s time period are shown in Table 8.

Next, we artificially changed the speed of the motor and chose five different speeds (A–E) between the maximum and minimum values. The STDs of vector geomagnetic measurements before and after AAIS within 2 s are shown in Table 9.

Finally, one or two motors were placed near the triaxial magnetometer to measure the geomagnetic field, and the STDs of the values before and after AAIS within an appropriate 2 s time period are shown in Table 10.

Comparing and analyzing the IR in Table 7, Table 8, Table 9 and Table 10, the results show that the inhibitory effect of AAIS on AMI is not associated with the relative direction, distance, rotation speed and quantity of the motors. Therefore, the algorithm has outstanding performance in practical applications.

It is worth emphasizing that AAIS is not sensitive to noise sources, but only to noise characteristics, including frequency, amplitude and phase. We have tested and reported numerous experiments with different AMIs, in order to simulate different noise signals. For example, the changes in motor speed are similar to changes in noise frequency; the changes in detection distance can be similar to changes in noise amplitude, etc. The results vary in that the AAIS algorithm has a good suppression effect on AMIs with different frequencies, amplitudes and phases; in other words, it works equally well for other types of noise sources.

## 6. Conclusions

The present research is the first to explore the AMI compensation method for geomagnetic vector measurement without reference sensors. This real-time compensation algorithm, AAIS, integrates the detection and elimination of AMI. When the frequencies of AMI do not overlap with the maneuver frequencies of the plane, the AAIS can eliminate AMI and retain maneuver information. Taking the leakage AMI from the DC brushless motor as an example, when the platform was static, the noise suppression ratio after AAIS in the three axes reached 88.4%, 67.2%, 92.6%, respectively; when the platform was dynamic, the noise suppression ratio reached 99.4%, 98.9%, 99.3%, respectively. In addition, the validity of the AAIS is not associated with the motor’s direction, distance, speed, or quantity. Therefore, the proposed algorithm is suitable for airborne, on-board, and shipboard geomagnetic vector detection. Future work on integrating it on an embedded platform to realize real-time compensation for AMI is planned.

## Figures and Tables

**Figure 1 sensors-22-03642-f001:**
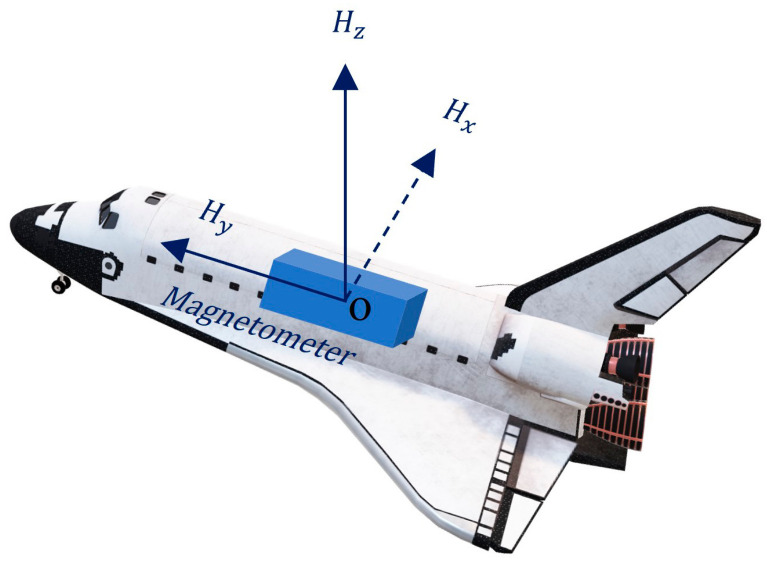
The magnetometer frame on a plane.

**Figure 2 sensors-22-03642-f002:**
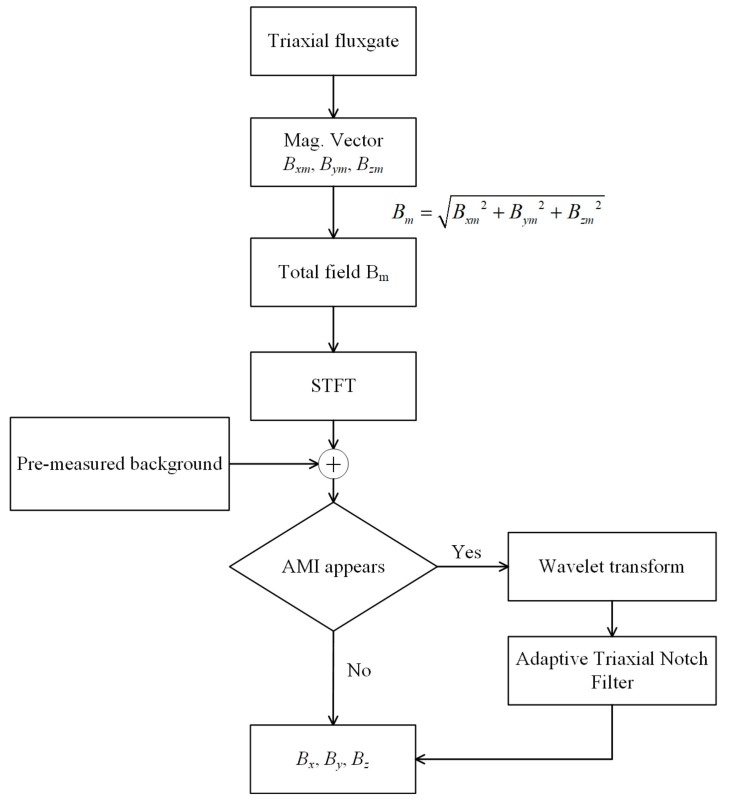
Block diagram of the AAIS algorithm.

**Figure 3 sensors-22-03642-f003:**
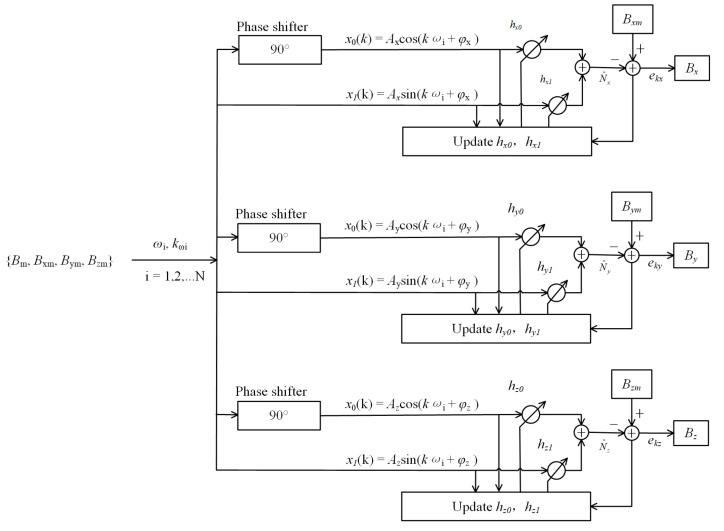
Schematic diagram of the triaxial adaptive notch filter.

**Figure 4 sensors-22-03642-f004:**
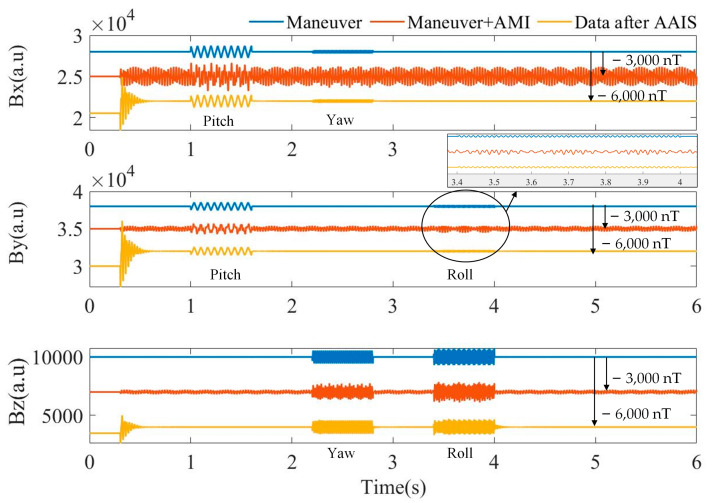
Simulation results. The components of a triaxial magnetometer in different situations. The blue curve shows the magnetic field change on each axis of the triaxial magnetometer, which is preset to simulate the plane maneuver. The orange curve shows the measurements of the triaxial magnetometer when AMIs exist. The yellow curve shows the data processed by AAIS.

**Figure 5 sensors-22-03642-f005:**
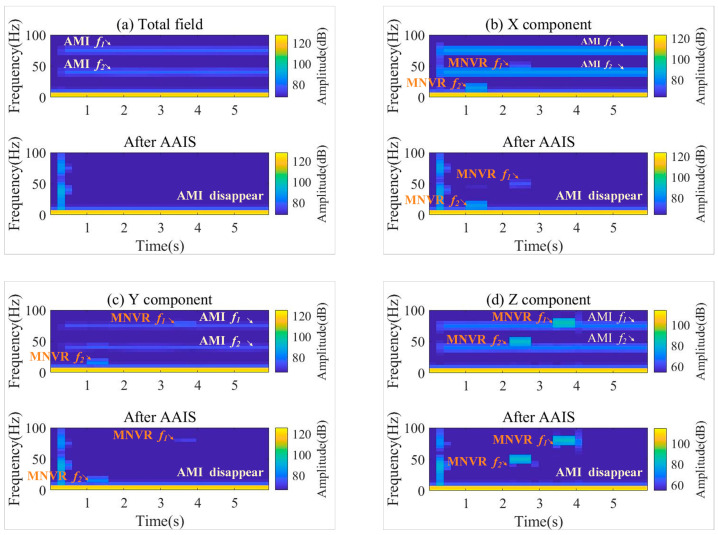
Time–frequency diagrams of the data before and after AAIS. (**a**) Total magnetic field; (**b**) X component; (**c**) Y component; (**d**) Z component. AMI is the abbreviation for alternating magnetic interference. MNVR is the abbreviation for maneuver. *f* means the frequency.

**Figure 6 sensors-22-03642-f006:**
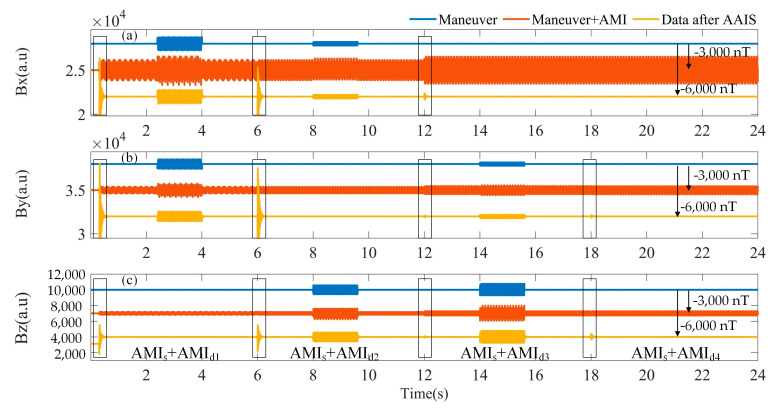
The suppression effect of AAIS on dynamic AMI. (**a**) X component; (**b**) Y component; (**c**) Z component. The blue curve shows the magnetic field change on each axis of the triaxial magnetometer, which is preset to simulate the plane maneuver. The orange curve shows the measurements of the triaxial magnetometer when AMIs exist. The yellow curve shows the data processed by AAIS.

**Figure 7 sensors-22-03642-f007:**
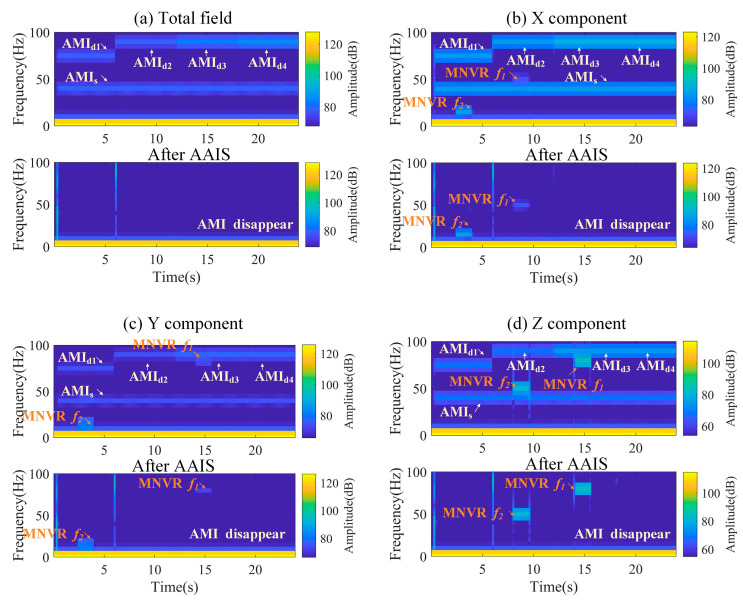
Time–frequency diagrams of the data before and after AAIS for the dynamic AMI. (**a**) Total magnetic field; (**b**) X component; (**c**) Y component; (**d**) Z component. AMI is the abbreviation for alternating magnetic interference. AMI_s_ is the abbreviation for static AMI. AMI_d_ is the abbreviation for dynamic AMI. MNVR is the abbreviation for maneuver.

**Figure 8 sensors-22-03642-f008:**
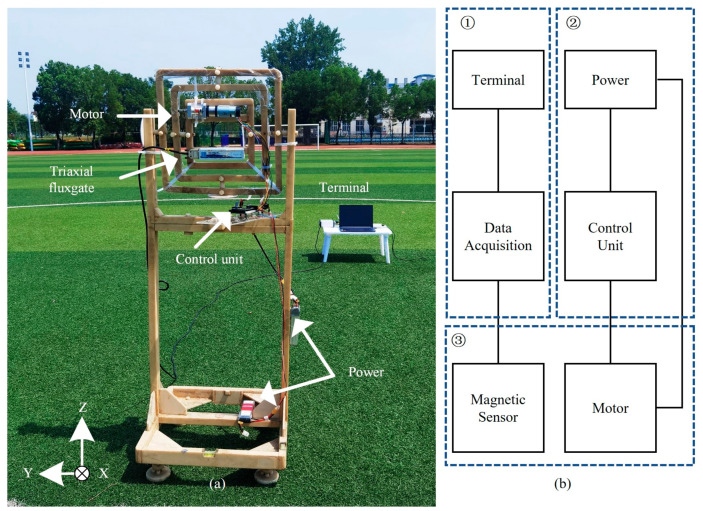
Experimental system and schematic diagram. (**a**) Experimental system; (**b**) Schematic diagram, including three parts: ① Data acquiring; ② Motor Control; ③ Magnetic field measurements.

**Figure 9 sensors-22-03642-f009:**
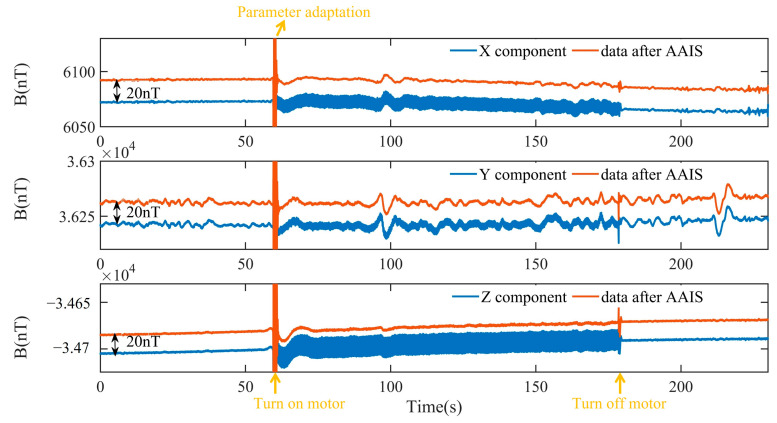
In static experiments, the components of the magnetic field before and after AAIS.

**Figure 10 sensors-22-03642-f010:**
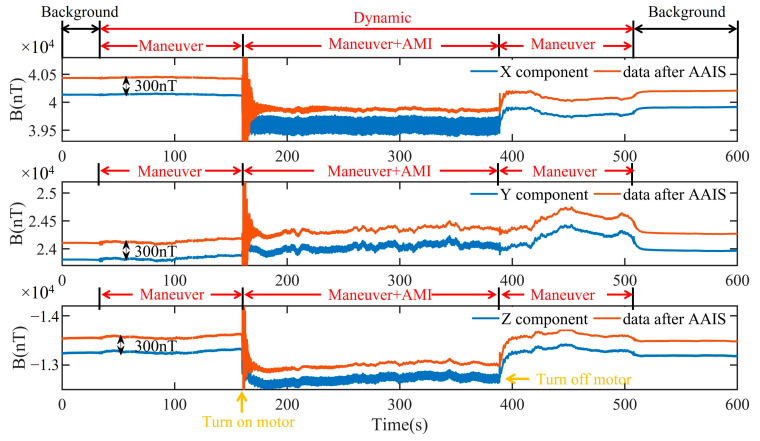
In the dynamic experiment, the components of the magnetic field before and after AAIS.

**Figure 11 sensors-22-03642-f011:**
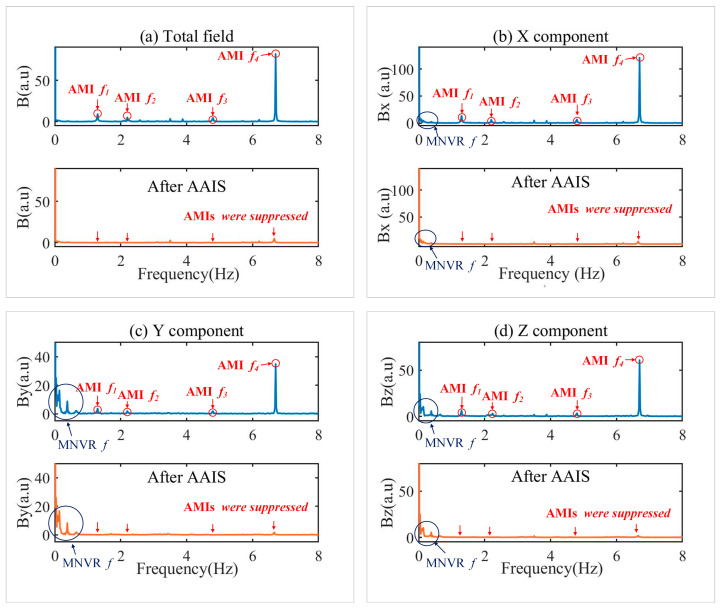
The frequency spectrum of the data measured by the magnetometer and processed by the AAIS algorithm. (**a**) The total magnetic field; (**b**) The X component; (**c**) The Y component; (**d**) The Z component. AMI is the abbreviation for alternating magnetic interference. MNVR is the abbreviation for maneuver. *f* means the frequency.

**Table 1 sensors-22-03642-t001:** The meaning of the characters.

Characters	
N	The number of the AMI frequency
$\omega_{i}$	The i-th value of the AMI angular frequency
$k_{\omega i}$	The discrete-time k when $\omega_{i}$ appears
$A_{x}$ $,A_{y}$ $\mathrm{and}\;A_{z}$	The amplitude of the AMI on the *x*, *y*, *z*-axes
$\varphi_{x}$ $,\;\varphi_{y}$ $\;\mathrm{and}\;\varphi_{z}$	The phase of the AMI on the *x*, *y*, *z*-axes
$x_{0}$ $\mathrm{and}\;x_{1}$	A set of orthogonal functions for estimating AMI
$h_{0}$ and $h_{1}$	The weight coefficient of the $x_{0}(k)$ and $x_{1}(k)$
${\hat{N}}_{x}$${\hat{N}}_{y}$ and ${\hat{N}}_{z}$	The estimated triaxial AMI at discrete time k
$e_{kx}$$,\;e_{ky}$ and $e_{kz}$	Residual error (MSE) after notch filter
$B_{x}$$,\;B_{y}$ and $B_{z}$	The magnetic field after notch filter

**Table 2 sensors-22-03642-t002:** The change in attitude of the plane.

	Frequency	Range	Start Time	End Time	Duration
pitch	15 Hz	±1°	1.0 s	1.5 s	0.5 s
roll	80 Hz	±1°	3.5 s	4.0 s	0.5 s
yaw	50 Hz	±1°	2.0 s	2.5 s	0.5 s

**Table 3 sensors-22-03642-t003:** Simulation values for dynamic AMI (AMI_d_).

	Duration	Frequency	Amplitude (x,y,z)	$\mathrm{Phase}\;(\varphi_{x},\varphi_{y},\varphi_{z})$
AMI_d1_	0~6 s	75 Hz	(600, 200, 100)	(0,π/2,π)
AMI_d2_	6 s~12 s	90 Hz	(600, 200, 100)	(0,π/2,π)
AMI_d3_	12 s~18 s	90 Hz	(1000, 300, 200)	(0,π/2,π)
AMI_d4_	18 s~24 s	90 Hz	(1000, 300, 200)	(0,π/3,π/6)

**Table 4 sensors-22-03642-t004:** The standard deviation (nT) for the dynamic AMIs.

Duration	Before AAIS			After AAIS		
	X	Y	Z	X	Y	Z
4.9~5.9 s	600.6	200.2	100.1	6.2 × 10^−10^	4.6 × 10^−9^	2.6 × 10^−11^
10.9 s~11.9 s	600.6	200.2	100.1	2.1 × 10^−10^	1.0 × 10^−10^	3.9 × 10^−11^
16.9 s~17.9 s	825.4	255.2	158.3	6.1 × 10^−10^	1.9 × 10^−10^	1.2 × 10^−10^
22.9 s~23.9 s	825.4	255.2	158.3	6.1 × 10^−10^	2.0 × 10^−10^	1.2 × 10^−10^

**Table 5 sensors-22-03642-t005:** The standard deviation (nT) in the static experiment.

	X	Y	Z
Before AAIS	3.63	1.83	7.31
After AAIS	0.42	0.6	0.54
IR	88.4%	67.2%	92.6%

**Table 6 sensors-22-03642-t006:** The standard deviation (nT) in the dynamic experiment.

	X	Y	Z
Before AAIS	7891.45	681.09	2062.70
After AAIS	50.38	7.23	14.10
IR	99.4%	98.9%	99.3%

**Table 7 sensors-22-03642-t007:** The standard deviation (nT) at different orientations.

Relative Position		West	North	East	South
X	Before AAIS	1315.35	8.59	302.33	17.78
	After AAIS	56.78	0.31	100.29	4.04
	IR	95.7%	96.4%	66.8%	77.3%
Y	Before AAIS	250.60	11.48	109.18	198.21
	After AAIS	10.11	2.53	13.04	58.06
	IR	96.0%	77.9%	88.1%	70.70%
Z	Before AAIS	163.75	15.00	226.99	289.42
	After AAIS	13.85	0.66	22.79	76.97
	IR	91.5%	95.6%	90.0%	73.4%

**Table 8 sensors-22-03642-t008:** The standard deviation (nT) at different distances.

Different Distance		5 cm	10 cm	15 cm
X	Before AAIS	27,862.24	356.57	189.57
	After AAIS	1512.90	19.45	8.44
	IR	94.6%	94.5%	95.5%
Y	Before AAIS	3272.10	1315.35	61.33
	After AAIS	203.74	56.78	3.37
	IR	93.8%	95.7%	94.5%
Z	Before AAIS	4097.93	163.75	13.04
	After AAIS	336.66	13.85	1.71
	IR	91.8%	91.5%	86.9%

**Table 9 sensors-22-03642-t009:** The standard deviation (nT) at different rotation speeds.

Different Speed		A	B	C	D	E
X	Before AAIS	356.57	741.43	605.62	601.68	859.57
	After AAIS	19.45	10.40	13.63	23.98	268.39
	IR	94.5%	98.6%	97.7%	96.0%	68.8%
Y	Before AAIS	1315.35	128.70	141.00	125.08	197.69
	After AAIS	56.78	3.52	3.64	4.73	64.09
	IR	95.7%	97.3%	97.4%	96.2%	67.6%
Z	Before AAIS	163.75	41.47	33.63	32.33	53.88
	After AAIS	13.85	3.78	1.78	2.87	12.84
	IR	91.5%	90.9%	94.7%	91.1%	76.2%

**Table 10 sensors-22-03642-t010:** The standard deviation (nT) with different motor numbers.

Different Numbers		X	Y	Z
one	Before AAIS	3.63	1.83	7.31
	After AAIS	0.42	0.6	0.54
	IR	88.4%	67.2%	92.6%
two	Before AAIS	55.73	18.05	15.74
	After AAIS	3.22	1.17	1.14
	IR	94.2%	93.5%	92.8%

## Data Availability

Not applicable.

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
