# Peer review of "An Adaptive Alternating Magnetic Interference Suppression (AAIS) Algorithm for Geomagnetic Vector Measurement"

_sensors, 2022, doi:10.3390/s22103642_

Round 1
Reviewer 1 Report
-Main contribution of the work is AAIS algorithm however this algorithm is tested assuming near to static magnetic noise. What would be impact of random noise generated during flight.
- How much error is introduced in measuring system by implementing the stated calibration approach
- Authors need to test and report more scenarios pertaining to dynamic nature of magnetic noise
- What about impact of ferromagnetic objects when placed in close proximity of sensing system
- The main assumption of magnetic noise in a flying object made in this paper needs to be revisited as the magnetic noise will be dynamic and not so straightforward to be modeled with a motor
Reviewer 2 Report
I think this is a good contribution to Geomagnetic Vector Measurement. There are some comments. More reference about the methods to overcome alternating magnetic interference (AMI) is needed. Is it appropriate using the DC brushless motor in the expriment to simulate the actual application? the speed scope, the power, the position, and maybe more than one motor are used in the aircraft.
Reviewer 3 Report
See the attachment.

Round 2
Reviewer 1 Report
Adequate changes have been made, thanks for addressing my concerns
Reviewer 3 Report
I think there is no problem.